# A Flexible 7-in-1 Microsensor Embedded in a Hydrogen/Vanadium Redox Battery for Real-Time Microscopic Measurements

**DOI:** 10.3390/membranes13010049

**Published:** 2022-12-30

**Authors:** Chi-Yuan Lee, Chia-Hung Chen, Yu-Chun Chen, Xin-Fu Jiang

**Affiliations:** 1Department of Mechanical Engineering, Yuan Ze Fuel Cell Center, Yuan Ze University, Taoyuan 32003, Taiwan; 2HOMYTECH Global Co., Ltd., Taoyuan 33464, Taiwan

**Keywords:** hydrogen/vanadium redox flow battery, micro-electro-mechanical systems (MEMS), high performance, corrosion-resistant flexible 7-in-1 microsensor, real-time microscopic diagnosis

## Abstract

The latest document indicates that the hydrogen/vanadium redox flow battery has better energy density and efficiency than the vanadium redox flow battery, as well as being low-cost and light-weight. In addition, the hydrogen, electrical conductivity, voltage, current, temperature, electrolyte flow, and runner pressure inside the hydrogen/vanadium redox flow battery can influence its performance and life. Therefore, this plan will try to step into the hydrogen/vanadium redox flow battery stack and improve the vanadium redox flow battery of this R&D team, whereof the electrolyte is likely to leak during assembling, and the strong acid corrosion environment is likely to age or fail the vanadium redox flow battery and microsensors. Micro-electro-mechanical systems (MEMS) are used, which are integrated with the flexible 7-in-1 microsensor, which is embedded in the hydrogen/vanadium redox flow battery for internal real-time microscopic sensing and diagnosis.

## 1. Introduction

In compliance with Taiwan’s energy-transformation policy, the government departments push renewable energy actively and plan to attain a long-term goal for net zero emissions in 2050. However, renewable energy is subject to the weather, leading to intermittent power generation and challenges to a stable power supply and maintaining power quality. Therefore, the Taiwan Power Company set up a power-trading platform to push ancillary service transactions and opened private enterprise investment in building energy-storage systems to provide ancillary services. In order to enhance the safety of energy-storage systems, the Ministry of Economic Affairs has stipulated special regulations for energy-storage systems referring to the consumer electrical appliance regulations of the national electrical safety code (NESC), according to which the users set up energy-storage systems, the employees design the construction, and the power companies inspect the power supply. Renewable energy is carbon-free and complements fossil fuels. Its advantage is that it reduces the value at risk, and its disadvantage is that it is expensive. Therefore, it can meet CO_2_ targets and a reasonable risk value at the same time. In the case of a reasonable risk value, the proportion of renewable energy power generation should be kept at about 25%. Especially after the realization of denuclearization in 2025, the proportion of renewable energy is a key power generation technology to support the national low-carbon goal and the low risk value of the power generation system. Gas-fired power generation has the advantage of low carbon, but the cost is high, and it can replace other fossil fuels, which has the disadvantage of increasing the risk value. Therefore, after denuclearization by 2025, although the proportion of power generation will increase significantly by 31% (due to additional low-carbon requirements), it can only maintain a proportion slightly above 30%. Coal-fired power generation is low-cost, but it has high CO_2_ emissions and is substituted with other fossil fuels, which is not good for increasing the value at risk. Therefore, after denuclearization by 2025, although the proportion of power generation will increase slightly by 39% (to supplement the demand for power supply security and low cost), with the increase in the proportion of renewable energy and gas-power generation, coal-fired power generation will be limited by high CO_2_ emissions, as hybrid generation will drop slightly to 38% [1,2]. In order to reduce the cost of the vanadium redox battery, Hsu et al. [3] substituted aqueous catholyte with gaseous hydrogen and maintained vanadium ions in the anolyte as a hydrogen/vanadium redox flow battery (HVRFB). The influence of the cathode was checked to study the HVRFB configuration, Pt load, humidity condition, and electrolyte flow velocity. The findings showed that the Pt load and the anolyte flow velocity had a significant impact on the electrolyte utilization. Yufit et al. [4] designed, assembled, and tested a novel regenerated cell based on an aqueous vanadium solution V(V)/V(IV) and hydrogen. The cell used gaseous anolyte and liquid catholyte to provide several latent advantages in large-scale application to the power grid. The charge and discharge experiments in different conditions showed encouraging reversibility and performance. Dewage et al. [5] indicated that placing reference electrodes in the redox flow battery without disturbing the battery operation was a significant challenge. The electrochemical impedance analyzer was used in the study of Dewage et al., and a substitute reference electrode setting was used to determine the losses of the cathode, the anode, and the whole battery. Their findings showed that the maximum irreversible loss came from the diffusion limitation in the cathode and the vanadium cross effect. These findings can further improve and optimize the cell design and material of two electrodes in the regenerative hydrogen/vanadium fuel cell. Jr et al. [6] used a Pt/C hydrogen electrode, a high-surface-area carbon nanotube (CNT) vanadium electrode, and an interdigitated flow-field plate at the two electrodes to assemble a hydrogen/vanadium redox battery. The half-cell performance was measured by using a reference electrode, and the film loss and vanadium cross effect on the hydrogen electrode were determined. Cyclic research was performed with a high current density of 300 mA/cm^2^, and the total energy efficiency was higher than 60%. Tenny et al. [7] found that the peak power performance of the hydrogen/vanadium reversible fuel cell with MWCNT carbon cloth electrode was 0.61 W·mg^−1^·cm^−2^, that of MWCNT carbon paper was 0.54 W·mg^−1^·cm^−2^, and that of common carbon cloth was 0.29 W·mg^−1^·cm^−2^. Jr et al. [8] observed higher performance in higher vanadium flow velocity, a thinner membrane, and the CNT vanadium electrode. The peak power density was higher than 540 mW/cm^2^ when the NR212 membrane and CNT vanadium electrode were used. Jr et al. [9] found that a thinner ion-exchange membrane increased the cross rate in comparison to commercial Nafion membrane, and manufacturing and testing the electrospinning mixed nano-fibrous membrane in the hydrogen/vanadium reversible fuel cell could reduce the cross rate and enhance the cell performance. Muñoz et al. [10] proposed a model allowed to simulate the system, and the regenerative fuel cell model was further founded. Garcia et al. [11] found that the hydrogen/vanadium system with high energy density was proved to have higher vanadium concentration, standard cell potential, and high theoretical storage capacity. The system was implemented by developing and using an HER/HOR catalyst; the catalyst had higher chemical stability for halogen containing electrolytes, and the conventional catalyst (Pt/C) was proved to degrade rapidly in experiments.

The experimental results of Hu et al. [12] showed that the Pt/C could be stable in electrolyte and maintain high catalytic activity in acidic conditions. In comparison to electrolysis, the electrolyte prepared using Pt/C showed similar cell performance (CE~93%, EE~85%), meaning the electrolyte prepared by using the catalytic reduction technique meets the quality requirement. Chakrabarti et al. [13] demonstrated that the reduced graphene oxide improved Freudenberg H23 carbon paper as an electrode in the regenerative hydrogen/vanadium fuel cell (RHVFC) for the first time. Under all of the current densities, the rGO-modified heat-treated CP resulted in better battery capacity utilization than the untreated CP. Pino-Muñoz et al. [14] studied the characteristic of 5 cm^2^ RHVFC; the characteristic was based on a lot of experimental measurements of vanadium electrolytes at different flow velocities. When the vanadium electrolyte and hydrogen flow was 100 mL·min^−1^, the polarization property maximum peak power density of the cell was 2840 W·m^−2^. Dung et al. [15] found that an increase in the thermal stability of the electrolyte with phosphate additive could be attributed to the formation of complexes between the additive and vanadium ions. The CV and EIS measurements showed that better electrochemical properties could be obtained by using phosphate additives. The activity of the electrolyte with additives was enhanced so that the VFB performance was enhanced. Therefore, the phosphate additive enhanced the long-term stability of V(V) electrolytes. Zhou et al. [16] proposed the ion concentration-dependent ionic mobility of VRFB and developed a 2D transient model that contained the proposed ionic mobility. This model can (i) estimate ionic conductivity more accurately; (ii) predict the cell voltage more accurately, especially with high current density; and (iii) simulate the concentration distribution and local current density in the electrode more truly. Tang et al. [17] used the Nernst equation to develop a dynamic model of cell potential; the result showed that under the default cell voltage limitation, the relationship between the capacity loss rate and the cycle index was influenced by the absolute and relative magnitudes of different vanadium ion diffusion coefficients. In the case without a side reaction, a higher diffusion rate is sometimes stabilized faster, and steady-state capacity is reached within fewer charge and discharge cycles. Rahman et al. [18] developed different additive formulas and performed a comprehensive evaluation, to reduce the thermal precipitation of a highly supersaturated V(V) solution in the application of VRFB as possible. Parz et al. [19] indicated that the critical electrode potential was not measured in the load-running investigation. After a period of short supply of hydrogen, the stoichiometric boom of hydrogen resulted in very high potential at the cathode electrode. This kind of influence is very important in the starting and terminal operating periods of hydrogen/vanadium fuel cell systems. Jiang et al. [20] proposed a strategy for enhancing the VRFB performance and a method to evaluate the battery performance. Their findings showed that the VRFB could implement 80.83% energy efficiency and 76.98% electrolyte utilization under 600 mA cm^−2^ high current density, and they provided a 2.78 W cm^−2^ high peak power density and ~7 A cm^−2^ limiting current density at room temperature. Additionally, the battery could circulate over 20,000 times steadily without apparent attenuation under 600 mA cm^−2^, demonstrating excellent cycle stability. Su et al. [21] found that the upstream channel of a serpentine interdigitated flow field was free of overflow. Assuming that the upstream channel has a higher pressure than the downstream channel, the liquid water will be easily pushed into the downstream channel. The number of serpentine interdigitated channels is smaller than that of the downstream interdigitated channels, so the gas volume velocity is high, and the liquid water is likely to be pushed to the outlet. The design of a serpentine interdigitated flow field seems more suitable for water removal, which has been proven in small flow fields, but the serpentine flow field generates better performance in larger active area batteries.

## 2. Development of the Flexible 7-in-1 Microsensor

This study successfully integrated seven sensing structures by using MEMS technology, including hydrogen, electrical conductivity, temperature, voltage, current, flow, and pressure. The process of the flexible 7-in-1 microsensor is shown in Figure 1.

(a)First of all, the PI film was cleaned with acetone and methanol organic solutions, respectively, and the residual methanol was removed by DI water. The surface dust and residual oil and fat were removed to enhance the adhesive ability of the thin film metal.(b)AZP4620 was coated, and the electrode patterns of micro hydrogen, electrical conductivity, temperature, pressure lower electrode, voltage, current, and flow sensors were defined by using photolithography.(c)Cr and Au were evaporated by an E-beam evaporator as the adhesion layer and sensing electrode layer.(d)LTC 9320 was coated as the protection layer after lift-off using acetone. The sensing areas and pins of micro voltage, current, and hydrogen sensors were defined by using the photolithography process so that they were exposed and not covered by the protection layer.(e)LTC 9305 was coated as a dielectric layer after the protection layer, and the sensing areas of micro-pressure were defined by using the photolithography process.(f)AZP 4620 was coated, and the upper electrode pattern of the micro pressure sensor was defined by using the photolithography process. Acetone was used for lift-off.(g)AZP4620 was coated, and the micro hydrogen sensor pattern was defined. The tin dioxide and Pt were evaporated on the micro hydrogen sensor, and acetone was used for lift-off. The production of a flexible 7-in-1 microsensor was completed. Figure 2 shows the optical micrograph of the flexible 7-in-1 microsensor.

## 3. Calibrations of the Flexible 7-in-1 Microsensor

The calibrations of micro temperature, pressure, flow, voltage, current, electrical conductivity, and hydrogen sensors were introduced in this section.

### 3.1. Temperature Calibration of Flexible 7-in-1 Microsensor

The flexible 7-in-1 microsensor and the thermometer of the BM-525 BRYMEN digital multimeter were put in a DENG YNG DS45 drying oven. After the reference control temperature was set up and stabilized, the resistivity of the micro temperature sensor was extracted. In the working temperature range, the resistivity was extracted at an interval of 5 °C. The micro temperature sensor was calibrated three times, and the mean of the measured calibration curve was taken. Figure 3 shows the average calibration curve of three micro-temperature sensors, and the curve approximates the linear variation.

### 3.2. Pressure Calibration of the Flexible 7-in-1 Microsensor

This study used a capacitive micro pressure sensor, which is a parallel plate capacitor of a sandwich structure. The capacitive micro pressure sensor contained a nonconducting dielectric layer sandwiched between two parallel electrodes, and a fixed pressure was applied to the micro pressure sensor by using a Druck-DPI 530 pressure controller. Meanwhile, the capacitance data were extracted by using a Wayne Kerr Electronics 4230 LCR meter. The calibration result is shown in Figure 4.

### 3.3. Flow Calibration of the Flexible 7-in-1 Microsensor

The micro flow sensor was mainly placed in the part of the liquid flow end because continuous fluid flow only existed in this part. The micro flow sensor was embedded in the hydrogen/vanadium redox battery to avoid the fluid inducing vibration and noise of the micro flow sensor. Afterwards, the power supply, micro flow sensor, and NI PXI 2575 were connected in series for measuring the current. The power supply supplied a steady voltage to the micro flow sensor and generated heat so that the current fell. The fluid carried the generated heat away, so that the current rose, and the flow was measured at this point. The vanadium redox was supplied by the YOTEC WS Series speed adjusting flow pump. The base of the water flow calibration range was 260 mL/min, and the measurement proceeded at an interval of 5 mL/min until 300 mL. The calibration result is shown in Figure 5.

### 3.4. Voltage, Current, and Electrical Conductivity Calibration of Flexible 7-in-1 Microsensor

This study used a digital multimeter to check whether the measurement of a micro voltage sensor and a micro current sensor is correct. The cell voltage and current were verified by using the electric meter in the calibration procedures, and then the micro sensing head was connected to the cell. Normal sensing of micro voltage and current sensors was verified if there was no error. The calibration is shown in Figure 6 and Figure 7. The micro electrical conductivity sensor was in normal operation.

### 3.5. Hydrogen Calibration of Flexible 7-in-1 Microsensor

The micro hydrogen sensor was calibrated by using the hydrogen and oxygen supplied from an eight-channel fuel-cell testing machine. The micro hydrogen sensor was embedded in the hydrogen/vanadium redox battery to make sure the space was filled with the currently tested gas, and then the sensor was linked with the NI PXI 2575 data-acquisition unit to capture the resistance changes. When the calibration began, the oxygen at a constant temperature and a constant flow was led in. The oxygen ions were adsorbed by the surface of the micro hydrogen sensor, and the resistance rose as the surface was full of oxygen ions. Then, the hydrogen at a constant temperature and a constant flow was led in. The hydrogen carried oxygen ions away from the surface of the micro hydrogen sensor so that the resistance dropped, and the hydrogen was tested by using the resistance difference, as shown in Figure 8.

## 4. Development of Hydrogen/Vanadium Redox Battery

The technology of this hydrogen/vanadium redox battery was built on a vanadium redox battery, so the materials employed were about the same. The only difference was that the vanadium fluid was replaced by hydrogen at one end.

### 4.1. Hydrogen/Vanadium Redox Battery Design

As half of the vanadium electrolyte was substituted by hydrogen, the weight of the hydrogen/vanadium redox battery was reduced significantly. In addition, the carbon felt that was in contact with the vanadium fluid (Figure 9) was replaced by carbon paper (Figure 10) as the hydrogen was applied, which is favorable for hydrogen reactions. Figure 11 is the stereogram of the hydrogen/vanadium redox battery.

### 4.2. Metal Collector Plate, End Plate, and Flow-Field Plate

The primary function of the collector plate was to collect the current in the hydrogen/vanadium redox battery through the bipolar plate and to connect the hydrogen/vanadium redox battery to the external load. The current from an external power supply was led to the bipolar plate during charging, of which the bipolar plate was made of stainless steel, as shown in Figure 12. The end plate provided a uniform pressure on the whole hydrogen/vanadium redox battery so that it was fully sealed. The eight small holes on four sides were designed for aligning the center during assembly. Iron rods were inserted in the holes to fix the center section during assembly in order to avoid dislocation, as shown in Figure 13. The channels of the flow-field plate were designed as four serpentine channels, as shown in Figure 14.

## 5. Flexible 7-in-1 Microsensor Embedded in Hydrogen/Vanadium Redox Battery

The developed flexible 7-in-1 microsensor was embedded in the hydrogen/vanadium redox battery for internal measurements. The flexible 7-in-1 microsensor was placed in the upstream, midstream, and downstream of the liquid flow end and hydrogen end, respectively, as shown in Figure 15.

### 5.1. Internal Voltage Measurement during Charging of Hydrogen/Vanadium Redox Battery

The internal voltage could be measured by connecting the micro voltage sensors at the liquid flow end and the hydrogen end. The measurement result is shown in Figure 16. The measured voltage is close to the value on the power supply, and the uptrends are identical, so the inside of the hydrogen/vanadium redox battery is in a stable electrochemical reaction state. The downstream voltage is greater than the midstream voltage and greater than the upstream voltage. It is speculated that the further downstream the vanadium electrolyte reacts, the longer the reaction time, so the higher the voltage.

### 5.2. Internal Current Measurement during Charging of Hydrogen/Vanadium Redox Battery

The hydrogen/vanadium redox battery was supplied with a steady constant current during charging, and the measured internal current was in the same steady state. It was induced that the internal electrochemical reaction was steady, as shown in Figure 17.

### 5.3. Internal Electrical Conductivity Measurement during Charging of Hydrogen/Vanadium Redox Battery

The electrical conductivity was the reciprocal of resistance, so the electrical conductivity in various regions inside the hydrogen/vanadium redox battery could be calculated from the voltage and current, as shown in Figure 18.

### 5.4. Internal Temperature Measurement during Charging of Hydrogen/Vanadium Redox Battery

Figure 19 is a graph showing the internal temperature measurement of the hydrogen/vanadium liquid flow. It was observed that the downstream warmed faster than the midstream and upstream, but the warming was close and finally leveled off.

### 5.5. Internal Flow Measurement of Hydrogen/Vanadium Redox Battery

The internal flow of the hydrogen/vanadium redox battery is shown in Figure 20. As the water inlet was under the oxygen end, the flow was relatively unsteady. The flow was relatively steady at the posterior inlet/outlet, but the flow velocity decreased due to the obstruction of the carbon felt.

### 5.6. Internal Pressure Measurement of Hydrogen/Vanadium Redox Battery

A fixed pressure was applied to the micro pressure sensor by using a Druck-DPI 530 pressure controller; meanwhile, the capacitance data were captured by a Wayne Kerr Electronics 4230 LCR meter, as shown in Figure 21.

### 5.7. Internal Hydrogen Measurement of Hydrogen/Vanadium Redox Battery

Figure 22 shows the internal hydrogen measurement of the hydrogen/vanadium redox battery. The hydrogen was admitted through eight channels in this study, and the measurement could be performed as long as the eight channels and battery were free of gas leakage or incomplete installation. The upstream micro hydrogen sensor responded first, followed by midstream and downstream. The interaction was intense at the beginning and gradually slowed down when the hydrogen molecules occupied the whole reaction area.

## 6. Conclusions

This study developed a flexible 7-in-1 microsensor by using MEMS technology successfully. This flexible 7-in-1 microsensor has seven sensing functions, is resistant to electrochemical environments, is available for real-time measurements, and can be placed in arbitrary positions inside hydrogen/vanadium redox batteries. The flexible 7-in-1 microsensor was embedded in the hydrogen/vanadium redox battery for the successful real-time microscopic measurement of seven important physical quantities and distribution.

## Figures and Tables

**Figure 1 membranes-13-00049-f001:**
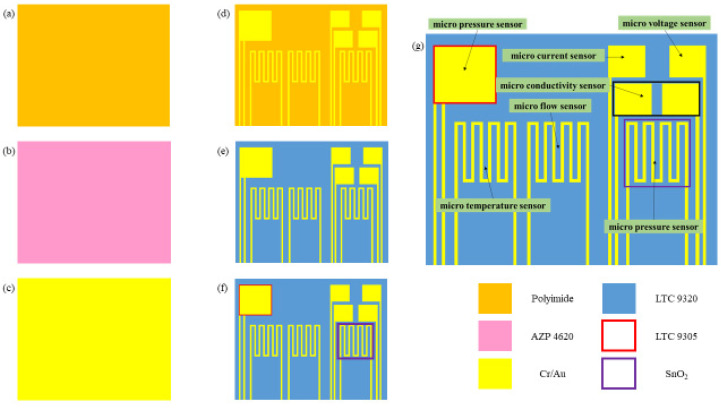
Process diagram of flexible 7-in-1 microsensor.

**Figure 2 membranes-13-00049-f002:**
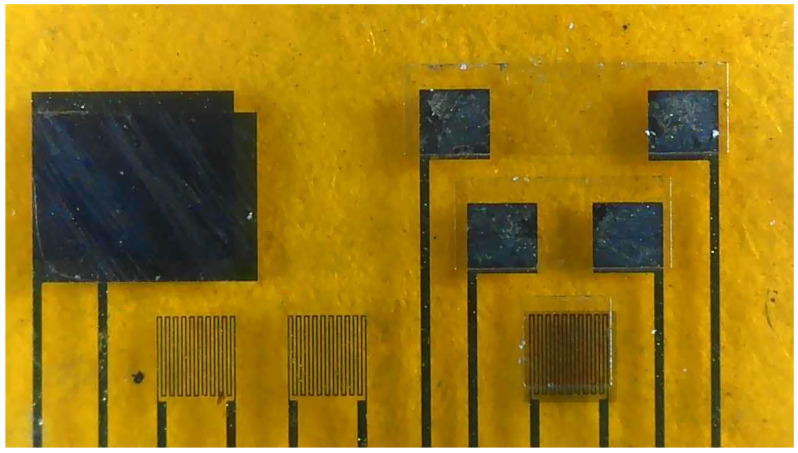
Optical micrograph of the flexible 7-in-1 microsensor.

**Figure 3 membranes-13-00049-f003:**
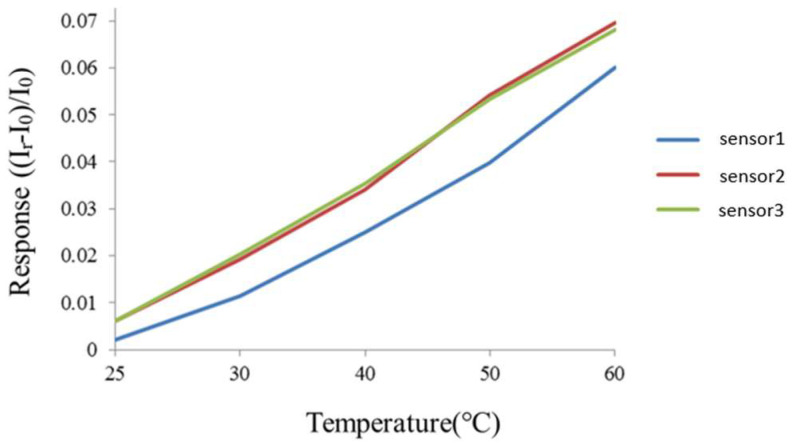
Correction curve of the micro temperature sensor.

**Figure 4 membranes-13-00049-f004:**
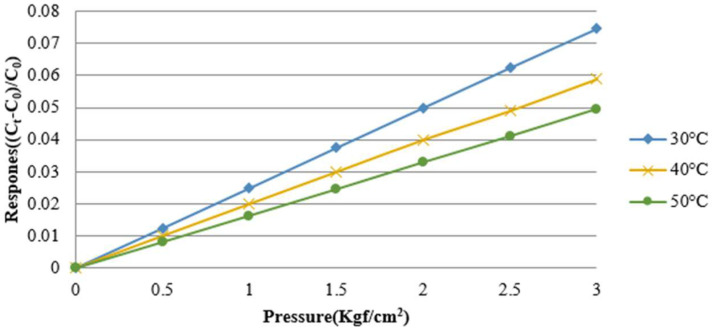
Correction curve of the micro pressure sensor.

**Figure 5 membranes-13-00049-f005:**
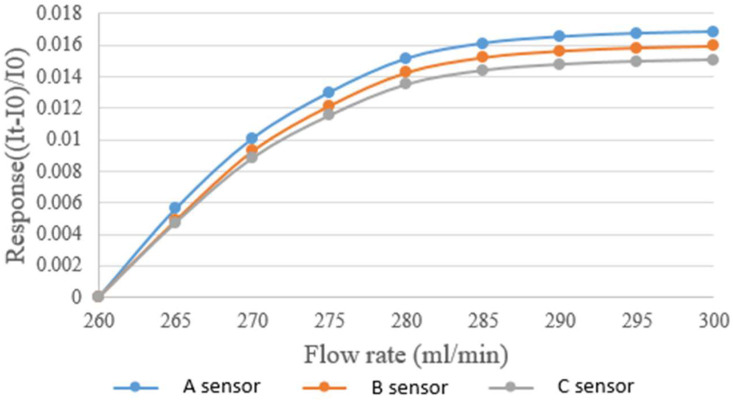
Correction curve of the micro flow sensor.

**Figure 6 membranes-13-00049-f006:**
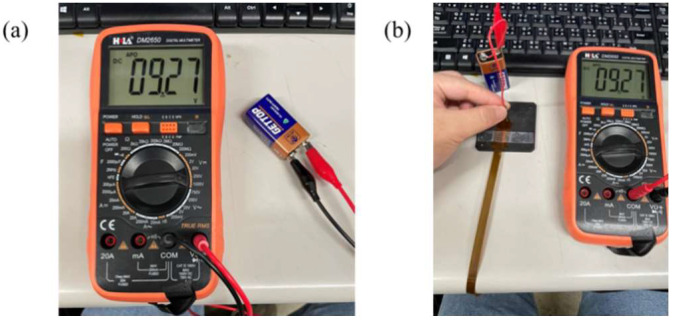
(**a**) Cell voltage measurement; (**b**) micro voltage sensor measurement cell.

**Figure 7 membranes-13-00049-f007:**
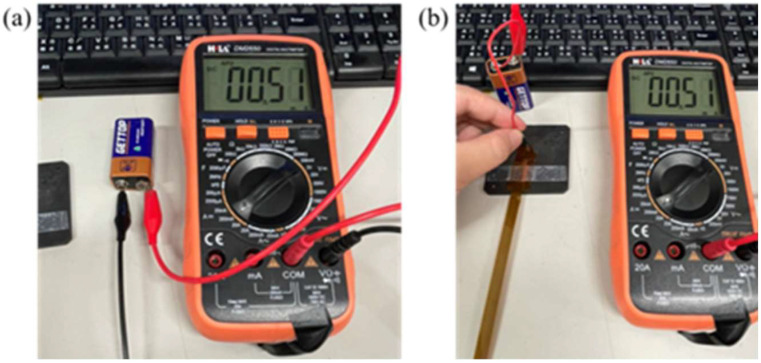
(**a**) Battery current measurement; (**b**) micro current sensor measurement cell.

**Figure 8 membranes-13-00049-f008:**
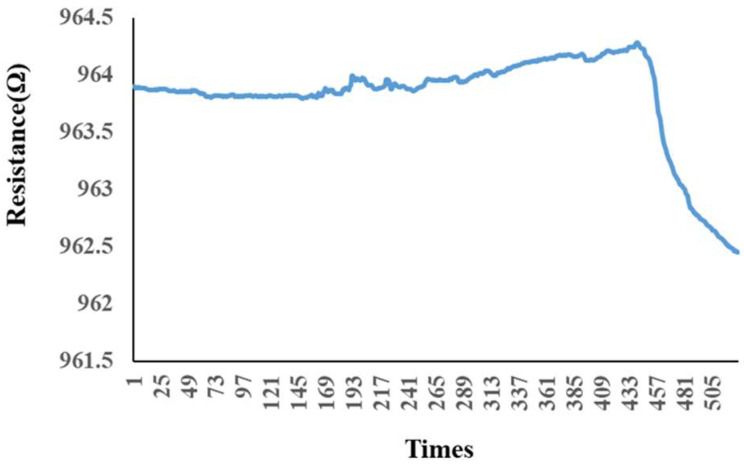
Variation of the micro hydrogen sensor after hydrogen was admitted and after oxygen was admitted.

**Figure 9 membranes-13-00049-f009:**
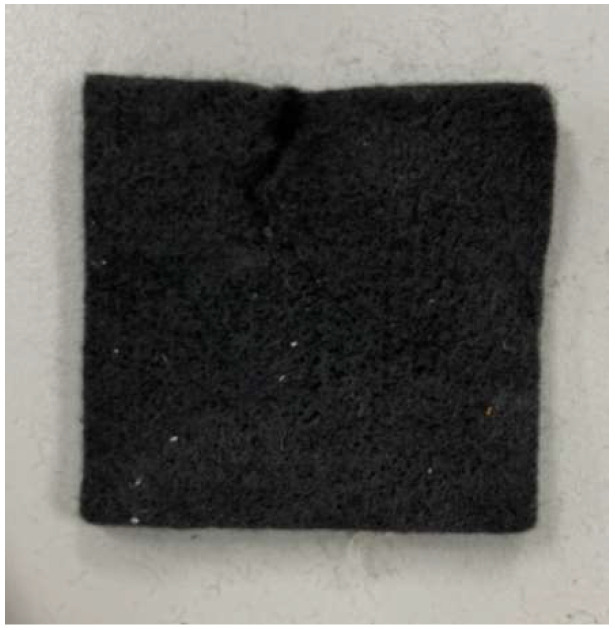
Carbon felt.

**Figure 10 membranes-13-00049-f010:**
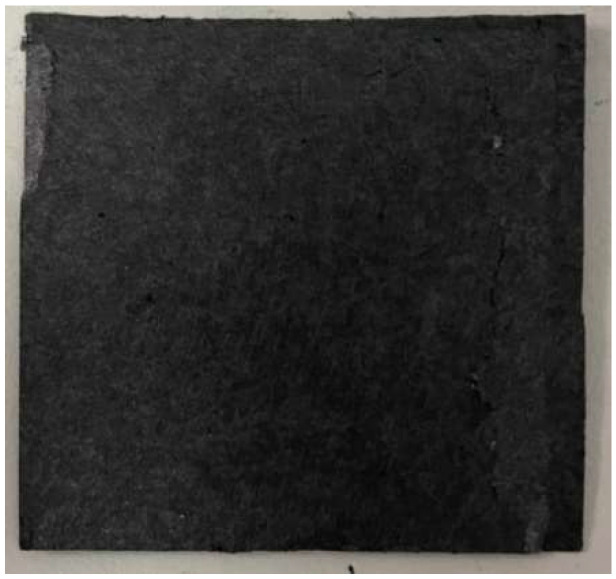
Carbon paper.

**Figure 11 membranes-13-00049-f011:**
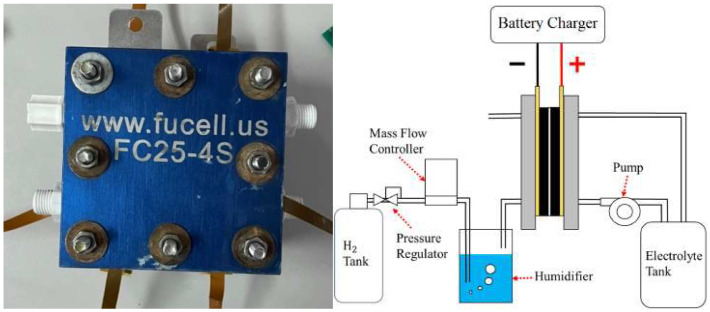
Stereogram of the hydrogen/vanadium redox battery and schematic diagram.

**Figure 12 membranes-13-00049-f012:**
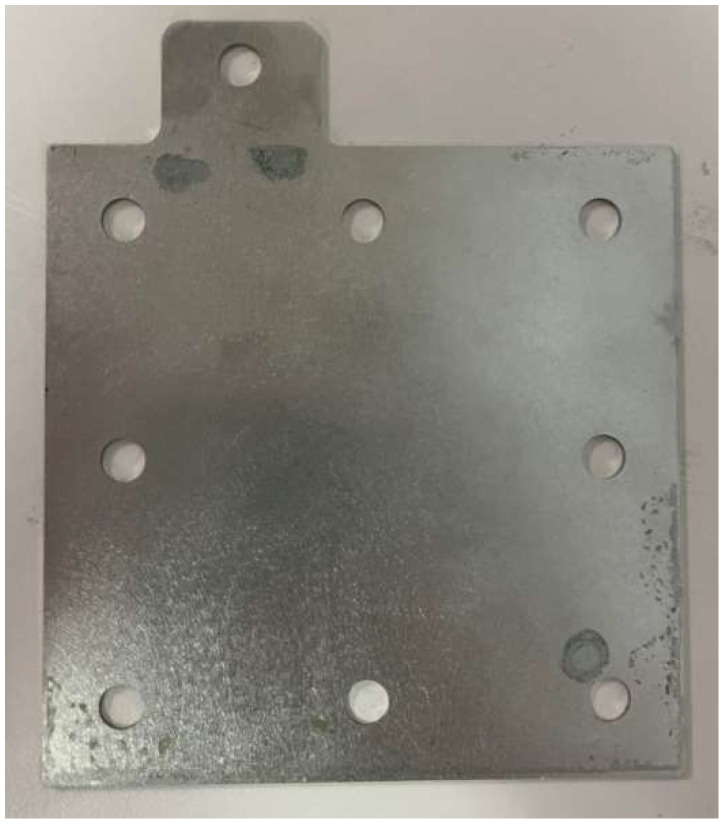
Metal collector plate.

**Figure 13 membranes-13-00049-f013:**
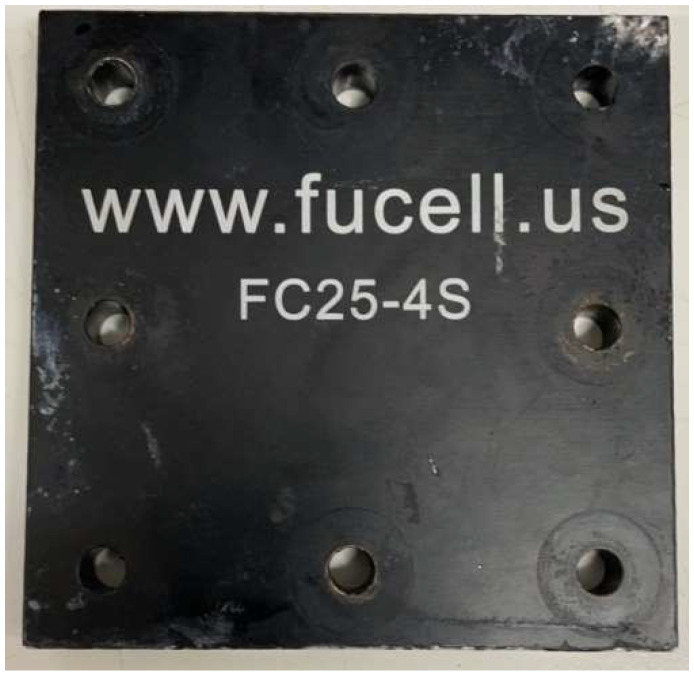
End plate.

**Figure 14 membranes-13-00049-f014:**
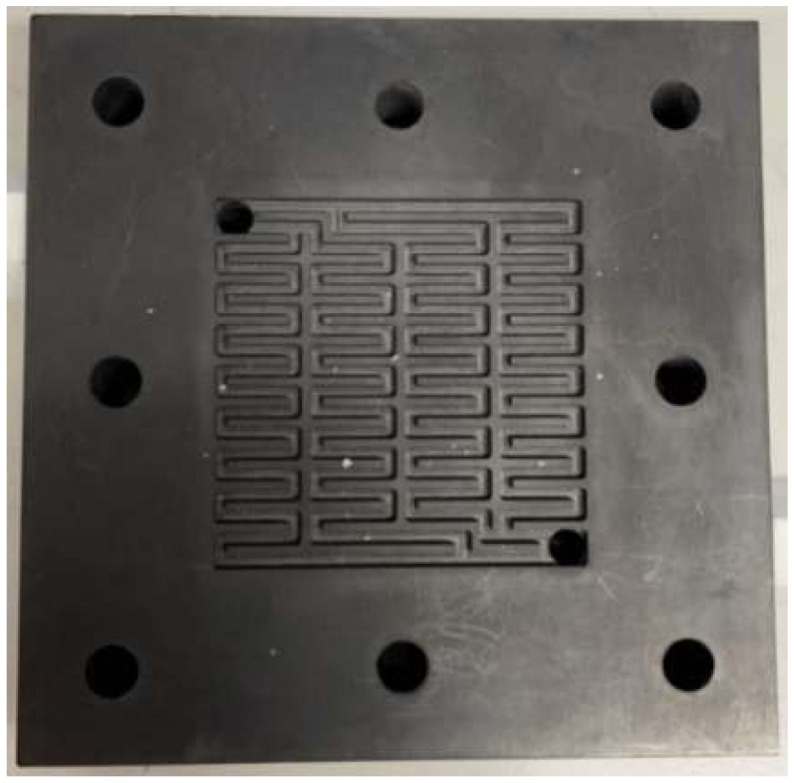
Flow field plate.

**Figure 15 membranes-13-00049-f015:**
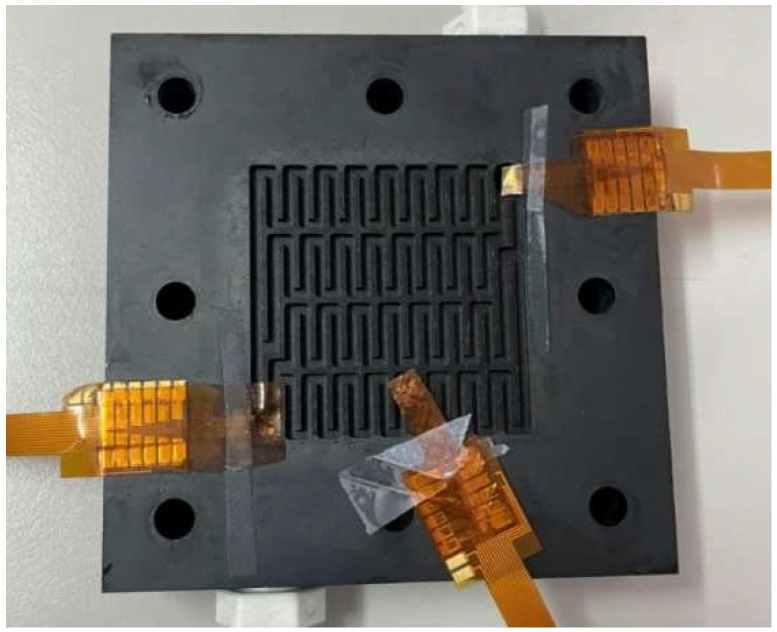
Position of the flexible 7-in-1 microsensor embedded in the hydrogen/vanadium redox battery.

**Figure 16 membranes-13-00049-f016:**
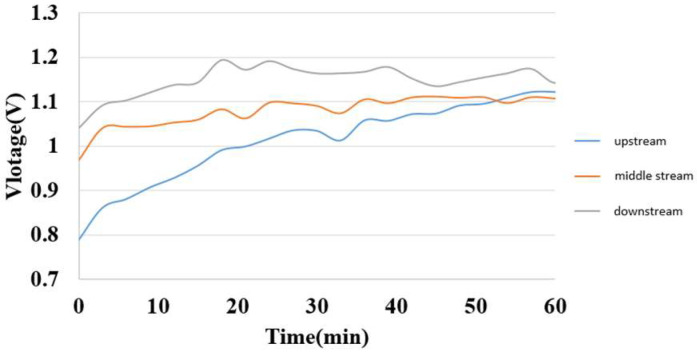
Internal voltage distribution during charging of hydrogen/vanadium liquid flow.

**Figure 17 membranes-13-00049-f017:**
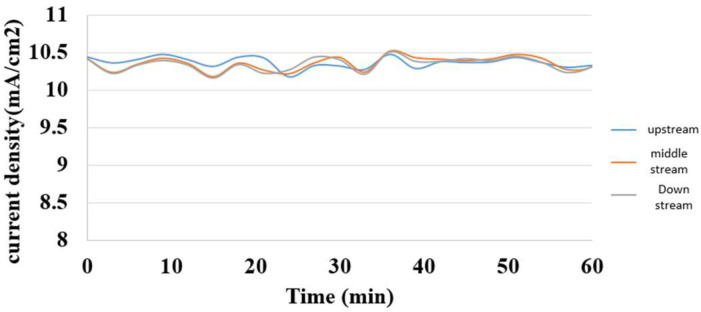
Internal current distribution during charging of the hydrogen/vanadium redox battery.

**Figure 18 membranes-13-00049-f018:**
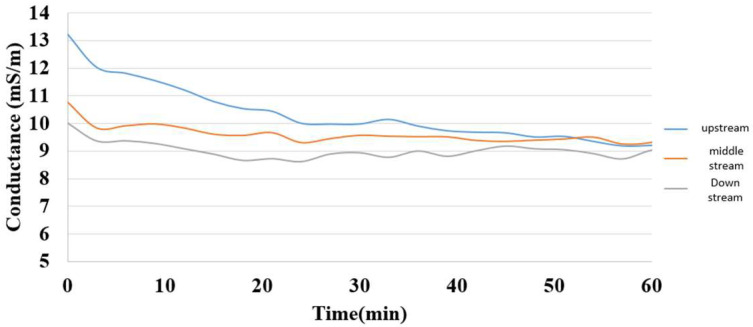
Internal electrical conductivity distribution during charging of the hydrogen/vanadium redox battery.

**Figure 19 membranes-13-00049-f019:**
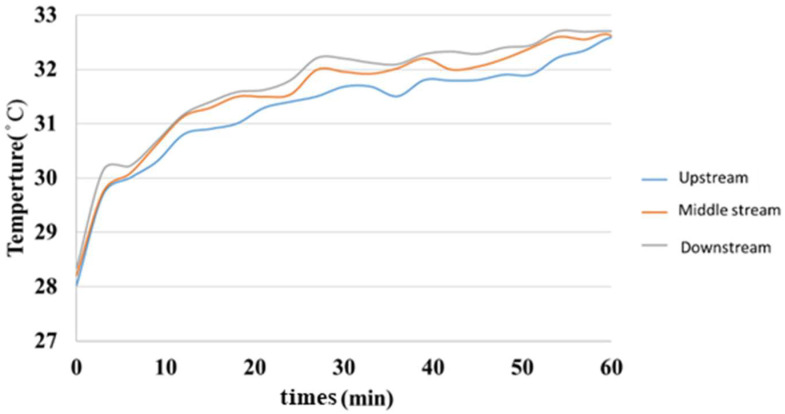
Internal temperature distribution of the hydrogen/vanadium redox battery.

**Figure 20 membranes-13-00049-f020:**
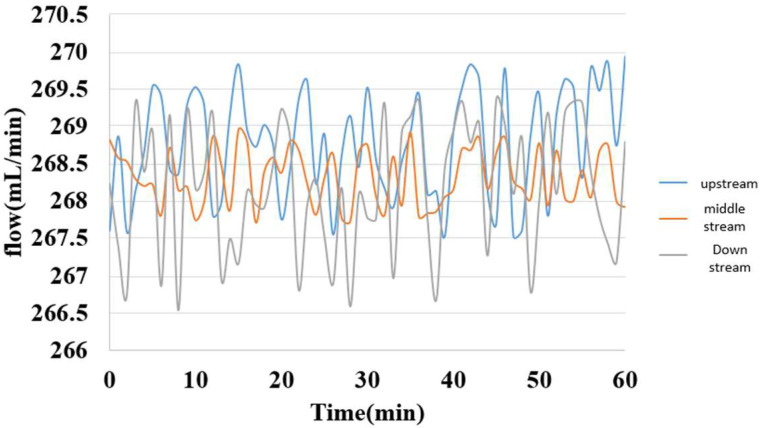
Internal flow distribution of the hydrogen/vanadium redox battery.

**Figure 21 membranes-13-00049-f021:**
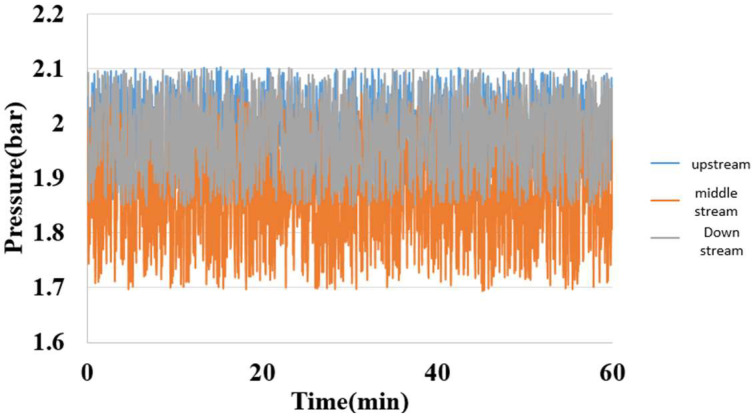
Internal pressure distribution of the hydrogen/vanadium redox battery.

**Figure 22 membranes-13-00049-f022:**
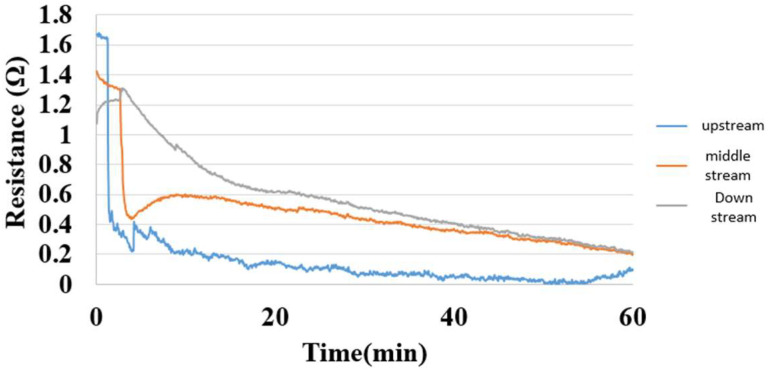
Hydrogen distribution inside the hydrogen end of the hydrogen/vanadium redox battery.

## Data Availability

Not applicable.

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
