# Peer review of "A Flexible 7-in-1 Microsensor Embedded in a Hydrogen/Vanadium Redox Battery for Real-Time Microscopic Measurements"

_membranes, 2022, doi:10.3390/membranes13010049_

Round 1
Reviewer 1 Report
In this work, a microsensor is embedded in the hydrogen/vanadium redox flow battery for internal real-time microscopic sensing and diagnosis. The important physical quantities and distribution of hydrogen/vanadium redox flow battery are successfully measured. Therefore, I suggest that this paper should be accepted for publish on Membranes after revision.
1. The schematic diagram of hydrogen/vanadium redox battery should be added to Fig. 11 for further clarify its internal structure.
2. The reasons of the internal voltage distribution has obvious difference in upstream, middle stream and downstream should be described.
3. Why is the upstream temperature higher than middle stream and downstream temperatures?
4. The efficiencies of hydrogen/vanadium redox battery should be provided in this manuscript.
5. I suggest that some figures should be combined. Moreover, the conclusion section can be further enriched.
Reviewer 2 Report
In this manuscript, the authors introduce a micro-electro-mechanical system with the flexible 7-in-1 microsensor, which can be used to detect the hydrogen, electrical conductivity, voltage, current, temperature, electrolyte flow and runner pressure of hydrogen/vanadium redox flow battery. In my opinion, this work is worthy of being published in Membranes.
Minor questions:
1. The process diagram in Figure 1 should be marked with numbers or letters to correspond to the following description.
2. Please confirm the photoresist type in process. it seems to be inconsistent with that in the process diagram.
3. How the microsensor is attached to the electrode? Can it remain stable under the flow of liquid and hydrogen?
4. The microsensor is fixed in the upper, middle, and lower positions of the electrode plate. In Figure 16, the voltage was measured by connecting sensors on both electrodes. Why are the voltage detected by the sensors different? Please provide the overall voltage and discharge voltage variation.
Reviewer 3 Report
1. In the abstract, in the beginning, it writes 'the hydrogen, electrical conductivity, voltage, current, temperature, electrolyte flow and runner pressure inside the hydrogen/vanadium redox flow battery', and later it writes 'the strong acid corrosion environment is likely to age or fail the vanadium redox flow battery and microsensors'. Is the research object of this paper VRFB or hydrogen-vanadium RFB? It's a little unclear here. It is recommended that the full text be consistent and unified.
2. 'micro-electro-mechanical systems (MEMS)' and 'high performance' should be included in the keyword.
3. On the first page, when referring to Taiwan's energy transformation policy, achieving zero net emissions by 2050, intermittent generation of renewable energy and other phenomena, it is recommended to add some references as theoretical support (Renewable and Sustainable Energy Reviews, 2007, 11(2): 345-356, Journal of Power Sources, 2021, 493: 229445, Energies, 2021, 14(15): 4402).
4. In the Introduction part, 'In order to reduce the cost of hydrogen/vanadium redox battery', this sentence appears suddenly. It belongs to energy storage methods, and energy storage should be introduced first. And then RFBs are very promising for long-term and stationary energy storage.
Moreover, VRFB is very well-known, why in this manuscript focus on hydrogen/vanadium RFB, not VRFB? Some basic working mechanisms and reasons should be given in detail for comparison (Electrochimica Acta, 2022, 408: 139937). And it should be 'in order to reduce the cost of VRFB', hydrogen/vanadium RFB is developed, not 'in order to reduce the cost of hydrogen/vanadium redox battery'.
Cost should not be the only reason, since VRFB does not require the loading of Pt, but in HVRFB there requires a Pt load. Hence, It seems that there is an advantage in cost for HVRFB compared to VRFB because half amount of vanadium is saved, but additional platinum needs to be added for HVRFB. It seems that there is an offset and compromise between the two systems. Further descriptions should be added to explain this issue.
5. On page 2, 'Their findings showed that the maximum irreversible loss came from the diffusion limitation in the cathode and the vanadium cross effect. These findings can further improve and optimize the cell design and material of two electrodes in the regenerative hydrogen/vanadium fuel cell.'
The correspondence of this sentence is inconsistent. For example, where does the vanadium cross-effect take place? It is due to the membrane material because the membrane is generally not an effective barrier layer for vanadium ions (Electrochimica Acta, 2021, 378: 138133). However, it is said later that electrode materials will be studied, which has little to do with vanadium ion crossover. In addition, this manuscript has been studying RFB. Why does the fuel cell appear here at the end of the sentence?
'Muñoz et al. [9] proposed a model allowed to simulate the system, and the regenerative fuel cell model was further founded. ' What is the so-called 'regenerative fuel cell', and what is the relation with HVRFB? It is quite confusing.
'The system was implemented by developing and using an HER/HOR catalyst, the catalyst had a higher chemical stability for halogen-containing electrolytes, and the conventional catalyst (Pt/C) was proved to degrade rapidly in experiments.' What is this HER/HOR catalyst exactly? It is clear that the conventional catalyst is Pt/C, but the new one is not very clear what it is. And in VRFB there is always sulfuric acid, so the halogen-containing electrolytes can be substituted easily to sulfuric acid to avoid degradation problems. What is the reason for halogen must be used in HVRFB? It should be explained.
6. The experimental process is almost unrelated to the membrane materials, more similar to the installation, calibration, adjustment and testing of sensors, as well as the construction of the entire battery system. Therefore, the relevance of the journal Membranes still needs to be further enhanced.
7. At present, the demonstration of the experimental results is sufficient, but the discussion of the results is very poor. It is suggested that the academic level of the discussion part of the results is not enough and should be further enhanced. Now the scientific soundness is limited due to the poor discussion part.
Round 2
Reviewer 1 Report
The revised manuscript has a publishing value after comprehensive and serious revision. Therefore, I suggest that the revised manuscript should be accepted for publication in Membranes.